# Sexual dysfunctions related to use of antipsychotics: A protocol for a systematic review and meta-analysis

Thalia Herder [1]*, Symen K. Spoelstra[2,3], Anuschka S. Niemeijer[4], Hendrikus Knegtering[5,6,7]

1 Department of Psychiatry, Martini Hospital, Groningen, The Netherlands, 2 Addiction Care North Netherlands, Groningen, The Netherlands, 3 NHL Stenden University of Applied Sciences, Leeuwarden, The Netherlands, 4 Scientific Institute, Martini Hospital, Groningen, The Netherlands, 5 Lentis Research, Lentis Mental Health Institute, Groningen, The Netherlands, 6 Department of Psychiatry and Rob Giel Research Center, University Medical Center Groningen, University of Groningen, Groningen, The Netherlands, 7 Mental Health Services GGz Drenthe, Department of psychotic illnesses, Assen, The Netherlands

* t.herder@mzh.nl

## Abstract

### Introduction

Sexual dysfunctions are a challenging side effect associated with antipsychotic treatment. This protocol outlines a systematic review and meta-analysis to assess the prevalence of overall sexual dysfunction, as well as the specific phases of sexual function affected by antipsychotic medications. Additionally, the analysis will explore the relationship between prolactin levels and sexual dysfunction.

### Methods and analysis

This protocol has been registered in the database of the International prospective register of systematic reviews (PROSPERO) under the registration number CRD42024573877. We will conduct a systematic search across electronic databases, including PubMed, MEDLINE, PsycINFO, Web of Science, the Cochrane Central Register of Controlled Trials (CENTRAL), Embase, and CINAHL, to identify relevant studies. Studies will be included if they meet predefined inclusion criteria, which include only controlled randomized trials assessing sexual functioning in patients receiving antipsychotic treatment. The antipsychotic medications of interest are amisulpride, aripiprazole, asenapine, brexpiprazole, cariprazine, chlorpromazine, clopenthixol, clozapine, droperidol, flupenthixol, fluphenazine, haloperidol, iloperidone, levomepromazine, loxapine, lurasidone, molindone, olanzapine, paliperidone, penfluridol, perphenazine, perazine, pimozide, prochlorperazine, quetiapine, risperidone, sertindole, sulpiride, thiothixene, thioridazine, trifluoperazine, ziprasidone, zuclopenthixol and zotepine. The primary outcome is the overall prevalence of sexual

**Data availability statement:** No datasets were generated or analysed during the current study. All relevant data from this study will be made available upon study completion.

**Funding:** The author(s) received no specific funding for this work.

**Competing interests:** The authors have declared that no competing interests exist.

dysfunction among patients undergoing antipsychotic treatment, while the secondary outcomes include the domains of sexual dysfunction (e.g., desire, arousal, orgasm) and serum prolactin levels. A network meta-analysis (NMA) will be performed using random effects to combine all available evidence for each outcome, aimed to provide a comprehensive ranking of different antipsychotics. NMA will be performed in R within a frequentist paradigm. The Cochrane Risk of Bias tool and RoB 2.0 will be used to assess the risk of bias in studies. We will evaluate the quality of the evidence contributing to network estimates for the primary outcomes using the GRADE framework, and key factors that may affect the observed effects will be analysed for consistency across studies.

## Introduction

Sexual dysfunction is believed to be highly prevalent among individuals with psychiatric disorders [1]. Numerous factors can significantly impact sexual health in patients with mental health disorders, including the disease or disorder itself, comorbidities, stigma or difficulties in engaging in social and sexual relationships [1,2]. Sexual dysfunction is a clinically significant impairment in sexual desire, arousal, orgasm, or satisfaction, as measured by validated instruments and reported by the patient as causing distress or functional impairment in the context of sexual activity [3].

In addition, specific psychotropic medications used to treat these disorders, such as antipsychotics, are known for their detrimental effects on sexual health [4]. In clinical practice, antipsychotics have a broad range of applications beyond schizophrenia. Antipsychotics are frequently prescribed for managing various other psychiatric disorders, such as bipolar disorder and major depressive disorder, as they are effective in alleviating symptoms and improving overall mental health [5–7]. The prevalence of sexual dysfunction among individuals using antipsychotic medication is estimated to range 16–60%, depending on the specific agent, assessment method, and population studied [8,9].

Sexual dysfunctions resulting from antipsychotic medications are recognized as most distressing side effects [10]. Sexual dysfunctions can lead to lower self-esteem, reduced quality of life, and reduced adherence to therapy [11,12]. Many patients with schizophrenia discontinue their antipsychotic treatment due to sexual dysfunction (42% of men and 15% of women) [13]. This discontinuation is linked to higher rates of relapse, rehospitalization, suicide mortality, cardiovascular mortality, and all-cause mortality [14].

Antipsychotics thought to affect the sexual response cycle directly through antagonistic effects on the dopamine system (e.g., desire, arousal, and orgasm) and indirectly contribute to sexual dysfunction (e.g., through weight gain, anticholinergic effects and elevated prolactin levels) [15,16]. The sexual side effects of antipsychotic medication can be explained through several neurobiological mechanisms. Although the group of antipsychotics is pharmacologically heterogeneous, they all have an affinity for the dopamine system, either as partial agonists or full dopamine antagonists, which are agents that block dopamine receptors—primarily D2 receptors—to

inhibit dopaminergic neurotransmission. The intensity of dopamine blockade and whether an antipsychotic is only blocking dopamine (antagonists) or also have their own dopaminergic activities (partial agonists) cause differential effects on the dopamine system and possibly the associated influence on sexual (dys)function. Dopamine plays a crucial role in various brain functions, including sexual interest, sexual arousal, orgasm, and learning. Manipulating dopaminergic receptors often affects sexual behaviour in mammals, including humans [17]. Consequently, many antipsychotics may impact sexual performance through dopaminergic mechanisms [18].

Prolactin secretion is regulated by the modulation of pituitary dopamine 2 (D2) receptors, with dopamine inhibiting prolactin secretion. Effects of antipsychotics on the dopamine receptors in the pituitary interferes with physiologic dopaminergic modulation, resulting in increased or decreased prolactin secretion in peripheral blood [5]. Hyperprolactinemia (abnormally elevated level of serum prolactin) can lead to hypogonadism, reduced testosterone levels, and disruption of the hypothalamic-pituitary-gonadal axis. Prolactin elevation as well as its secondary disruption of the hypothalamic-pituitary-gonadal axis is thought to contribute to decreased sexual desire, sexual arousal, anorgasmia, amenorrhea, infertility, gynecomastia, and galactorrhoea [19]. However, newer antipsychotics with partial agonistic effects on the dopamine system may hardly influence or even decrease prolactin secretion.

Other neurotransmitter receptors that may contribute to effects of antipsychotics on sexual functioning are histaminergic, serotonergic, cholinergic, and alpha-adrenergic receptors. These receptor systems may impact sexual function by inhibiting for example motivation and reward, as well as peripheral bodily functions (e.g., peripheral vasodilation) [20].

Newer antipsychotics, including aripiprazole, brexpiprazole, cariprazine, and lurasidone, have been introduced for possible beneficial side effect profiles and potential advancements in the treatment of affective disorders [21]. Partial dopamine agonistic effects and/or serotonergic agonistic effects on the 5HT1a receptors may improve initiative, reduce anxiety and improve depressive symptoms. Some studies found that these agents are indeed less likely to cause sexual dysfunction [22]. Third generation antipsychotics (aripiprazole, brexpiprazole and cariprazine) are all partial dopaminergic agonists) [23]. They have been primarily developed aiming to cause less movement disorders and improve negative symptoms [24,25]. They are less sedating and having only a limited influence on prolactin levels in comparison to first- and second-generation antipsychotics [5]. Some studies suggest that these partial dopamine antagonists are less associated with undesired effects on sexual functioning [10]. Lurasidone, acts as an antagonist at D(2) and 5-HT [7] receptors and as a partial agonist at the 5-HT(1A) receptor subtype [26]. These newer antipsychotics may hold promise in offering dual benefits: efficacy in treating of psychotic and affective symptoms while minimizing the likelihood of causing sexual dysfunction.

In 2011, a meta-analysis on sexual dysfunction in patients with mental health disorders using antipsychotics was performed and showed that different antipsychotics influence various dimensions of sexual function [8]. Since then, significant advancements have occurred, including the approval and widespread use of newer antipsychotic agents, alongside a growing body of research investigating their side effect profiles. Given these developments, there is a clear need to update and expand upon the previous synthesis [27]. This updated review differs from the 2011 meta-analysis by incorporating recent studies, including newer antipsychotics, and by broadening the scope to include biological correlates such as prolactin. Furthermore, our meta-analysis is based exclusively on controlled studies, enhancing methodological rigor and minimizing bias. By focusing on randomized controlled trials and other controlled designs, this review aims to provide more robust and clinically relevant estimates of antipsychotic-induced sexual dysfunction.

## Aim

The aim of this meta-analysis is to systematically review and quantitatively analyse the available literature on sexual functioning associated with antipsychotic use. We will perform network meta-analysis techniques, which integrate both direct evidence from clinical trials and indirect evidence, facilitating comparisons across all antipsychotics and generating rankings based on their effects on overall sexual functioning and specific domains of the sexual response cycle. This approach will present available evidence into an accessible format, empowering patients and healthcare providers to make

informed, evidence-based treatment decisions to promote shared decision making. Tailored treatment with antipsychotics may contribute to improve sexual health and overall well-being. Additionally, the systematic review may also identify gaps in research that may inspire future studies.

## Methods and analyses

We will perform a systematic review and meta-analysis according to Preferred Reporting Items for Systematic Reviews and Meta-Analyses (PRISMA-P) guidelines (see S1 File PRISMA-P). Prior to start of article inclusion, study methods is registered in an international prospective register of systematic reviews (PROSPERO) protocol, registration number: CRD42024573877.

### Search strategy

Various databases, including PubMed, MEDLINE, PsychINFO, Web of Science, Cochrane Central Register of Controlled trials (CENTRAL), Embase and CINAHL will be searched. For published, unpublished and ongoing RCTs, a manual search will be conducted utilizing the following trial registries and websites: the European Medicines Agency (EMA) in the European Union, the Food and Drug Administration (FDA) in the USA, the Medicines and Healthcare products Regulatory Agency (MHRA) in the UK, the Medicines Evaluation Board (MEB) in the Netherlands, the Medical Products Agency (MPA) in Sweden, the Pharmaceuticals and Medical Devices Agency (PMDA) in Japan, and the Therapeutic Goods Administration (TGA) in Australia. There will be no language restrictions, and the search will encompass articles from their inception up to 1 September 2024 and will be re-run on 1 July 2025. For eligible articles published in languages unfamiliar to the reviewers, machine translation tools (e.g., ChatGPT 4.0 or newer versions) will be employed, complemented by back-translation or secondary human verification of critical methodological sections to ensure accuracy and preserve the integrity of the data. The search will be conducted by a professional librarian.

### Study design

Only blinded and open randomized controlled trials will be included. Open-label randomized trials will be included in this meta-analysis to ensure comprehensive coverage of the available evidence, particularly given the limited number of double-blind randomized controlled trials assessing sexual dysfunction in patients treated with antipsychotics. In the case of a crossover study, only the first phase will be considered for data-extraction. All other study designs will be excluded.

### Inclusion criteria

Studies will be included if they involve patients diagnosed with a primary mental health disorder, defined as the foremost, clinically significant psychiatric condition currently affecting the individual's functioning and constituting the main focus of diagnosis or treatment, requiring antipsychotic medication. This includes psychotic disorders (including schizophrenia), depressive disorders, bipolar disorder, obsessive-compulsive disorder, anxiety disorders, and post-traumatic stress disorder (PTSD), diagnosed according to DSM-III(-R), DSM-IV(-TR), DSM-5(-TR), ICD-10, or ICD-11 criteria. Both antipsychotic-naive patients and patients with a history of antipsychotic use will be included, in order to capture a representative clinical population and reflect real-world prescribing practices. Furthermore, the included patients in these studies must be 18 years or older and both female and male patients are included.

### Exclusion criteria

Studies will exclude patients diagnosed with concomitant somatic diseases known to cause sexual dysfunction or difficulty (e.g., neurological conditions, diabetes mellitus, thyroid disease, renal disease, hypogonadism, androgen disorders, etc.), as well as patients using multiple antipsychotic agents. Studies involving postpartum patients will also be excluded, as this condition can significantly impact sexual function [28]. Substance use is not considered an exclusion criterion, as

excluding these patients would significantly limit the generalizability of findings and is often not feasible due to high comorbidity rates in clinical populations treated with antipsychotics [29].

## Types of intervention

We will include controlled studies on sexual (dys)functions and, if also available, serum prolactin levels in antipsychotics registered in Europe or the USA (i.e., amisulpiride, chlorpromazine, clopenthixol, droperidol, fluphentixol, fluphenazine, haloperidol, loxapine, molindone, perphenazine, pimozide, prochlorperazine, thiothixene, thioridazine, trifluoperazine, aripiprazole, asenapine, brexpiprazole, cariprazine, clozapine, iloperidone, levomepromazine, lurasidone, olanzapine, paliperidone, penfluridol, perazine, quetiapine, risperidone, sulpiride, sertindole, ziprasidone, zuclopenthixol, zotepine). Outcome measures for sexual functioning will include (MeSH) terms such as sexual functioning or performance, sexual desire (disorders), sexual arousal (disorders), and orgasmic (disorders) along with related terms (see supplementary S1 Appendix) [2,5].

## Outcomes

Studies reporting on the prevalence of sexual dysfunctions in psychiatric patients using antipsychotics and also assessing sexual dysfunction by standardized and validated sexual questionnaires will be included, e.g., PRSexDQ (Psychotropic-Related Sexual Dysfunction Questionnaire), ASEX (Arizona Sexual Experience Scale), UKU (Udvalg for Kliniske Undersøgelser), CSFQ (Changes in Sexual Function Questionnaire), ASFQ, or any other validated and standardized rating scale [30]. Studies reporting sexual dysfunction rates based on spontaneous reports or semi-structured interviews lacking validated and standardized assessment tools will be excluded. This approach ensures the use of validated and standardized questionnaires, thereby enhancing the internal validity of the analysis and minimizing heterogeneity across included studies, which is essential for robust quantitative synthesis and comparability. Studies combining two or more outcome measures (e.g., combining desire and arousal disorders) or rating sexual dysfunctions that does not align with standard outcome measures will also be excluded.

If eligible studies are included, they will be screened for the potential quantification of serum prolactin levels. If available, changes of serum prolactin levels will be measured from baseline to week 8 (or as close to 8 weeks as possible). We will use the internationally standardized normal values of serum prolactin [31].

## Primary outcome

The rate of sexual dysfunctions in patients after 8 weeks (or as close to 8 weeks as possible) of medication use, will be assessed as overall sexual dysfunction (either as continuous or dichotomous outcome). This outcome will be assessed by using the reported scores from validated sexual function rating scales [32]. We will report the number and percentage of patients reporting sexual dysfunction per study arm at 8 weeks (or as close as possible). If scores at 8 weeks are not reported but change scores are available, we will utilize the latter scores.

## Secondary outcomes

1. The rate of three aspects of sexual dysfunction will be studied: desire, arousal (erection and lubrication), orgasm (including delayed and premature ejaculation in men) [33]
2. The rate of changes in serum prolactin levels

## Identification and selection of studies

Two independent reviewers will screen and select articles through a systematic process. Titles, abstracts and full-text screening will be evaluated independently by both reviewers. In cases of uncertainty or disagreement, a third reviewer

will be consulted to make a final decision regarding inclusion. The selected studies will be screened for potential cross-references. Registration studies for the included antipsychotics will be reviewed to identify any unpublished data on sexual function. In case of multiple articles based on the same dataset, only the primary article will be included. A detailed record of the reasons for excluding articles will be created (see Fig 1). An online systematic review program (Covidence) will be used to facilitate a structured identification and selection process.

**Study timeline**

This study follows a structured approach, progressing through key stages of data collection and analysis. As of 11 February 2025, we are in the early phases of the study, specifically focused on record screening. The timeline for the remaining phases is estimated as follows:

Record Screening: currently ongoing, with an expected completion by 1 June 2025.

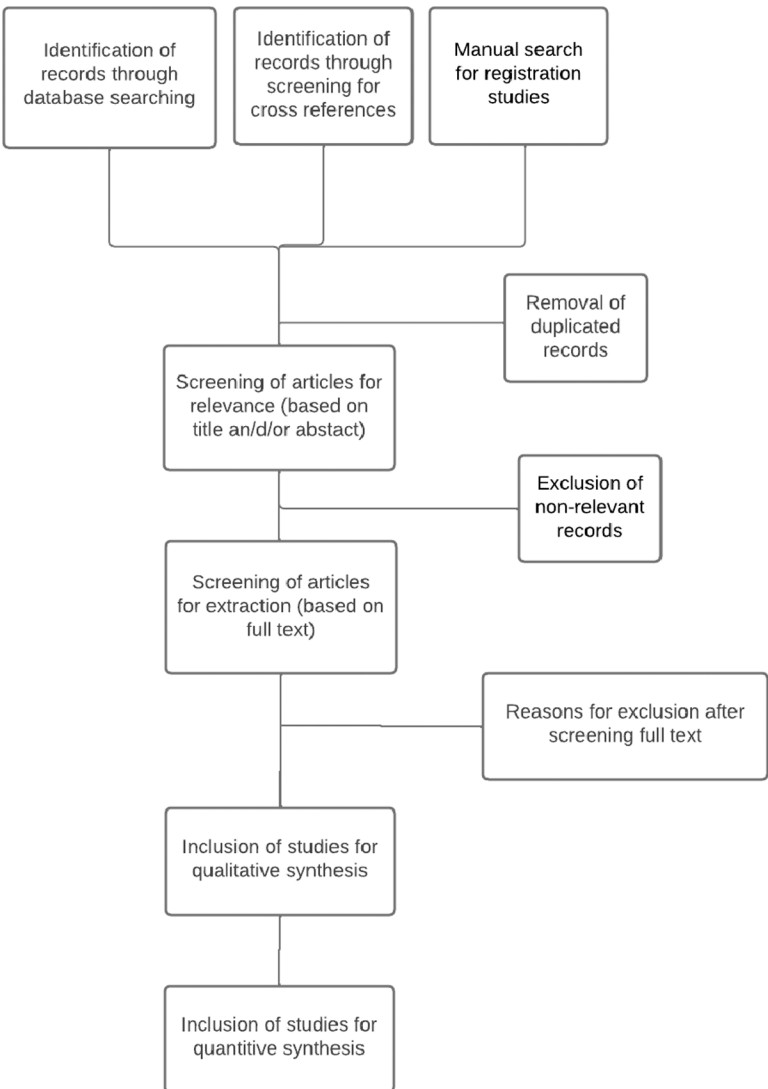

**Fig 1. Decision-tree for selecting articles and extracting data.**

Data Extraction: scheduled to begin following the completion of record screening, estimated to be completed by 1 August 2025.

Results Analysis and Reporting: Once data extraction is finalized, analysis will commence, with preliminary results expected by 1 September 2025.

## Data extraction process

Two independent reviewers will extract data from the selected articles. To ensure accuracy, they will cross-verify the input of data into the final dataset. In addition, study authors will be contacted for clarification on unclear data or to provide any missing information.

By using a standardized data extraction form, all relevant data per study will be collected, including:

- Study characteristics: Information such as the first author, year of publication, journal, sample size, country where the study was conducted, study design (e.g., non-blinded), name of the questionnaire used to assess sexual functioning, trial duration (follow-up), and sponsorship.

- Participant demographics: Data on participant sex (% female), mean age, diagnosis, baseline prolactin levels, baseline sexual dysfunction, and use of concomitant antidepressants.

- Details of antipsychotic medication: Information on the antipsychotic treatments studied in each study arm, including the name of the medication (or placebo), dosage (flexible or fixed), dosage comparability (yes/no), and duration of use.

- Outcomes per study arm: Key outcomes such as the number of participants randomized, number (%) of patients reporting sexual dysfunction, mean scores with standard deviations on sexual function questionnaires, number of missing data points, number lost to follow-up, number of imputed data points, imputation techniques used, and serum prolactin levels.

Data will be extracted from an 8-week follow-up, as this period aligns with most trial designs for evaluating sexual function after starting antipsychotics. If 8-week data are unavailable, outcomes closest to the 8-week mark will be used. Both fixed-dose and flexible-dose designs will be included. When comparing drugs at different dosage levels within their therapeutic ranges (e.g., upper versus lower limits), a dichotomous variable is used to indicate dosage comparability [34].

## Measurement of (adverse) effects on sexual function

We will use the number and percentage of patients reporting sexual dysfunction per study arm at 8 weeks (or the closest available time point). If scores at 8 weeks are unreported but change scores are provided, these will be used instead.

When the number of patients experiencing sexual dysfunction is not directly reported but means (with SD or SE) for baseline and endpoint data on a sexual dysfunction rating scale are available, Furukawa et al.'s method will be applied to estimate the number of affected patients [32]. Furukawa et al found that assuming normal distributions, while often such symptom measures are skewed and not normally distributed, did not affect the conclusions drawn. We will perform sensitivity analyses to assess the impact of including studies in the analyses in which data was imputed.

When response rates are imputed from means and SDs, a decrease of 50% or more in baseline scores will indicate sexual dysfunction in the absence of established norms. The SD will be used to estimate variability, with SEs converted into SDs. If neither SDs nor SEs are available, they will be calculated using confidence intervals, *t*-values, or *p*-values, following the methods outlined in the Cochrane Handbook for Systematic Reviews [35,36]. If studies present the results with and without imputation of missing data, imputed data will be prioritized, and completer-only analyses will be disregarded.

Mixed models will be preferred over the Last Observation Carried Forward (LOCF) method, as they provide a more accurate and reliable approach to handling missing data by incorporating all available information and accounting for variability over time [37]. If data are presented in graphical form only, values will be extracted from the figures. When other methods are unfeasible, the original study authors will be contacted for clarification or additional information.

## Risk of bias assessment

The Cochrane Risk of Bias (RoB 2.0) tool for randomized controlled trials will be used to assess the quality of included studies [38]. This evaluation will be conducted independently by two reviewers. Any discrepancies between reviewers will be resolved through consensus, with the involvement of an additional member of the review team for resolution if necessary. When insufficient information is provided, we may contact the authors to request additional details.

We will address the potential biases in the following areas: the generating of the allocation sequence, concealing the allocation, blinding of both study personnel and participants, blinding of outcome reviewers, management of attrition, selective reporting of outcomes, imputation techniques used, and other relevant factors, including potential sponsorship bias. When insufficient information about allocation concealment and other trial characteristics is supplied, it may be necessary to reach out to the authors of the trial for additional details.

The quality assessment for bias risk will be categorized in three categories: low risk of bias (none of the evaluated aspects were considered as high risk, and three or less were considered as unclear risk), moderate risk of bias (one of the above evaluated aspects was considered as high risk, or none were considered as high risk, but four or more were rated as unclear risk) and high risk of bias (all other scores that do not fit the criteria for low or moderate risk of bias).

## Statistical analysis

**Descriptive statistics.** We will compile descriptive statistics to summarize the trial and study population characteristics from all eligible trials. This process will involve outlining the various types of comparisons and outline of key clinical and methodological variables, including year of publication, type of psychiatric disorder, proportion of participants allocated to placebo, age, sponsorship, and clinical setting. This comprehensive overview will offer valuable insights into the diversity and scope of the included studies.

**Pair-wise meta-analysis.** Initially, we will conduct a traditional pairwise meta-analyses by pooling studies that evaluate identical interventions (antipsychotics), including comparisons between active treatments and various control groups. We will use a random-effects frequentist model for these pairwise meta-analyses.

We will quantify heterogeneity using tau-squared, which represents the between-study variance. We will assume a common heterogeneity variance across treatment comparisons and classify the level of heterogeneity as low, moderate, or high using empirical distributions based on the first and third quantiles [39,40]. Additionally, we will explore whether treatment effects for the two primary outcomes are robust in subgroup analyses and network meta-regression using the following study and participant characteristics: [1] publication year, [2] psychiatric diagnosis, [3] industry sponsorship, [4] treatment duration, and [5] gender distribution. The sensitivity of our findings will be further evaluated by conducting analyses restricted to: [1] studies with complete outcome data (i.e., no imputation performed by the primary authors), [2] studies in which we did not impute response data based on means and SD's, [3] studies with comparable antipsychotic dosages across arms, [4] studies providing unpublished data, [5] studies assessed as having a low risk of bias according to the predefined risk of bias criteria, [6] blinded trials, and [7] direct head-to-head trials only.

To address potential multicollinearity among covariates in the network meta-regression analyses, we will examine variance inflation factors (VIFs). Variables exhibiting high collinearity (i.e., VIF values exceeding acceptable thresholds) will be either excluded from the model or combined with related variables, as appropriate, to preserve model stability and interpretability.

**Network meta-analysis.** The fundamental assumption of network meta-analysis (NMA), known as transitivity, will undergo thorough analysis. A comprehensive evaluation of treatments will assess whether the included studies are comparable, as significant intransitivity can result in misleading conclusions. To validate indirect evidence, clinical and methodological variables serving as potential effect modifiers (e.g., sex, age, dosage) will be examined across various treatment comparisons [41]. This analysis will determine whether these variables are equally distributed among studies

within each comparison category. We will focus on: sex, mean age, diagnosis, baseline prolactin levels, baseline sexual dysfunction, use of concomitant antidepressants, dosage of anti-psychotic medication, type of dosage (flexible or fixed), and duration of use.

If the gathered studies demonstrate sufficient transitivity, a random-effects network meta-analysis (NMA) will be conducted to synthesize both direct and indirect evidence for outcomes related to sexual dysfunction and prolactin levels. The results will generate ranking of all treatments, presented as summary risk ratios (RRs) or standardized mean differences (SMD) in a league table, complete with confidence intervals. We will visually represent the available direct evidence using a network diagram. The size of each node in the diagram will reflect the volume of evidence for each treatment, based on the total number of patients involved. The connecting lines will be weighted according to the inverse of the variance of the summary effect from each comparison. If the conditions for network meta-analysis are not fulfilled, only the results of the pairwise syntheses will be reported.

We expect substantial heterogeneity between studies, which may lead to inconsistency. We will apply the separate indirect from direct evidence (SIDE) method, to assess the agreement between indirect and direct evidence for every possible comparison in the network [42]. In case we observe local inconsistency, we will examine possible sources like data-entry mistakes or differences in study characteristics.

To ensure the robustness of different medical treatments on our outcomes, we will conduct subgroup analyses based on the following characteristics: [1] baseline levels of sexual function, [2] blinding, [3] type of psychiatric disorder, [4] concomitant with antidepressants, [5] gender and [6] age. In addition, we will assess the validity of our conclusions by re-analysing the following conditions:

- Including only studies with reported standard deviations (SD) rather than imputed values.

- Including only studies with balanced doses in all arms (excluding studies with unequal dose comparisons).

- Including only studies with a low risk of bias.

As the potential for selection bias is elevated in trials involving antipsychotics, we will evaluate whether there are differences in outcomes between less precise trials and more precise trials [43]. Comparison-adjusted and contour-enhanced funnel plots are generated when more than 10 studies are included. To further explore associations, network meta-regression models will be used to examine the relationship between study size and effect size. If a significant association is identified, publication bias is suspected, we will investigate funnel plot asymmetry using a selection model to account for potential publication bias.

The R-package 'netmeta' will be employed for all these analyses. In addition, within R, we will use the netsplit (SIDE method) and the RoB 2.0 tool to evaluate the risk of bias due to missing evidence in the NMA [44]. We will consider $p$-values $< .05$ as statistically significant.

## GRADE

The review will conclude with a comprehensive summary of findings and an evaluation of the confidence in the evidence. Using the Grading of Recommendations Assessment, Development, and Evaluation (GRADE) framework, we will evaluate the overall certainty of evidence for each outcome. Results will be categorized and presented in outcome tables, in accordance with The Cochrane Collaboration's recommendations [45].

## Ethics and dissemination

Ethical approval is not required for this systematic review and meta-analysis as we perform a secondary analysis of publicly available data. Upon completion, the review will be submitted for publication in a peer-reviewed journal. The dataset and other supplementary materials will be made available. Also, the results of this review will be disseminated through print, social media, and presentations at relevant conferences.

## Discussion

By systematically comparing the sexual side effects and prolactin-related impacts of various antipsychotics, this study aims to provide clinicians with evidence-based guidance for selecting medications that minimize sexual dysfunction, a key factor affecting patient adherence. Improved understanding of these side effect profiles can enhance shared decision-making, support personalized treatment plans, and ultimately improve treatment adherence and quality of life [2]. Furthermore, the findings may inform the development of targeted interventions and monitoring strategies to mitigate sexual dysfunction, thereby optimizing long-term psychiatric care.

The protocol's rigorous methodology ensures reliable and valid findings, with strengths in its broad scope, advanced analytical approach, and high practical relevance. The review evaluates a wide range of antipsychotics, including newer-generation options, and employs validated sexual function questionnaires to ensure standardized outcome measures. Incorporating prolactin levels as a secondary outcome, the study provides insight into potential mechanisms linking antipsychotic induced alterations of prolactin levels to sexual dysfunction. Network meta-analysis facilitates treatment comparison and ranking, supporting evidence-based clinical decision making. Furthermore, exploration of heterogeneity through subgroup analyses, meta-regression, and robustness checks enhances depth and credibility of the findings. Clinically, the findings will guide shared decision-making, identifying less harmful options, ultimately improve adherence, reduce relapse, and enhance patient outcomes.

Potential challenges include heterogeneity in study designs, populations, and dosages, which may introduce confounding despite the use of subgroup analyses and meta-regression. Bias in included trials, such as inadequate blinding or pharmaceutical sponsorship, will be mitigated using the Cochrane Risk of Bias 2.0 tool, though complete elimination of bias is unlikely. The use of subjective self-reported questionnaires, although validated, may introduce recall bias. Consequently, some nuances related to patient-reported experiences and context-specific factors might not be fully captured, which could limit the comprehensiveness of the findings. Another limitation of this review is the exclusion of studies that rely on spontaneous reports or semi-structured interviews without the use of validated and standardized instruments for assessing the prevalence of sexual dysfunction. While this decision enhances the internal validity and comparability of included studies by reducing heterogeneity, it may also result in the omission of potentially valuable qualitative data and clinically relevant insights. Publication bias, where studies remain unpublished, will be addressed using funnel plots and selection models. Additionally, differences in participant characteristics across studies may challenge the assumption of transitivity in network meta-analysis, requiring cautious interpretation of results.

By proactively addressing these limitations, the study aims to provide clinically relevant findings to improve the management of sexual side effects in patients treated with antipsychotics, ultimately enhancing patient care and quality of life.

## Conclusion

The findings of this meta-analysis can directly inform clinical practice guidelines by providing robust, comparative evidence on the sexual side effect profiles of various antipsychotic medications. Incorporating these results into treatment recommendations will enable clinicians to better balance efficacy and tolerability when selecting antipsychotics, particularly for patients at risk of or experiencing sexual dysfunction. This evidence-based approach supports personalized medicine by guiding prescribers toward options with lower sexual adverse effects, potentially enhancing patient adherence and satisfaction. Furthermore, the integration of these findings into clinical guidelines can promote shared decision-making between clinicians and patients, fostering informed discussions about treatment choices and potential side effects. Ultimately, this will contribute to improved long-term outcomes by reducing treatment discontinuation related to sexual dysfunction and enhancing overall quality of life.

## Supporting information

**S1 File.** **PRISMA-P (Preferred Reporting Items for Systematic review and Meta-Analysis Protocols) 2015 checklist.**
(PDF)

**S1 Appendix.** **Search terms for study screening.**
(PDF)

## Acknowledgments

The authors thank Margot Verleg (MV) for her expertise and assistance in conducting the search for articles.

## Author contributions

**Conceptualization:** Thalia Herder, Symen K. Spoelstra, Hendrikus Knegtering.

**Data curation:** Thalia Herder, Anuschka S. Niemeijer.

**Formal analysis:** Thalia Herder, Anuschka S. Niemeijer.

**Investigation:** Thalia Herder, Anuschka S. Niemeijer.

**Methodology:** Thalia Herder, Symen K. Spoelstra, Anuschka S. Niemeijer, Hendrikus Knegtering.

**Project administration:** Thalia Herder, Symen K. Spoelstra, Hendrikus Knegtering.

**Resources:** Thalia Herder, Anuschka S. Niemeijer.

**Software:** Thalia Herder, Anuschka S. Niemeijer.

**Supervision:** Symen K. Spoelstra, Hendrikus Knegtering.

**Validation:** Thalia Herder, Anuschka S. Niemeijer.

**Visualization:** Thalia Herder, Symen K. Spoelstra, Anuschka S. Niemeijer, Hendrikus Knegtering.

**Writing – original draft:** Thalia Herder, Symen K. Spoelstra, Hendrikus Knegtering.

**Writing – review & editing:** Thalia Herder, Symen K. Spoelstra, Hendrikus Knegtering.

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
