## [Decision Letter · Decision Letter 0]

19 May 2025

PONE-D-25-05627Sexual difficulty related to use of antipsychotics: a protocol for systematic review and meta-analysisPLOS ONE

Dear Dr. Herder,

Thank you for submitting your manuscript to PLOS ONE. After careful consideration, we feel that it has merit but does not fully meet PLOS ONE’s publication criteria as it currently stands. Therefore, we invite you to submit a revised version of the manuscript that addresses the points raised during the review process.

We look forward to receiving your revised manuscript.

Kind regards,

Hidetaka Hamasaki

Academic Editor

PLOS ONE

Journal Requirements:

Reviewers' comments:

Reviewer's Responses to Questions

**Comments to the Author**

1. Does the manuscript provide a valid rationale for the proposed study, with clearly identified and justified research questions?

Reviewer #1: Yes

Reviewer #2: Yes

Reviewer #3: Yes

Reviewer #4: Yes

2. Is the protocol technically sound and planned in a manner that will lead to a meaningful outcome and allow testing the stated hypotheses?

Reviewer #1: Partly

Reviewer #2: Yes

Reviewer #3: Yes

Reviewer #4: Partly

3. Is the methodology feasible and described in sufficient detail to allow the work to be replicable?

Reviewer #1: Yes

Reviewer #2: Yes

Reviewer #3: Yes

Reviewer #4: Yes

4. Have the authors described where all data underlying the findings will be made available when the study is complete?

Reviewer #1: Yes

Reviewer #2: Yes

Reviewer #3: Yes

Reviewer #4: Yes

5. Is the manuscript presented in an intelligible fashion and written in standard English?

Reviewer #1: Yes

Reviewer #2: Yes

Reviewer #3: Yes

Reviewer #4: Yes

6. Review Comments to the Author

You may also provide optional suggestions and comments to authors that they might find helpful in planning their study.

Reviewer #1: This manuscript presents a protocol for a systematic review and meta-analysis investigating the association between antipsychotic use and sexual dysfunction. Given the significant impact of sexual dysfunction on treatment adherence and quality of life, this study aims to provide a comprehensive synthesis of existing evidence, comparing different antipsychotic medications.

Below are my specific comments:

1. Inclusion of Open-Label Trials

• The inclusion of open-label trials raises concerns about potential bias, particularly given the subjective nature of sexual dysfunction assessments. Unblinded studies may lead to over- or underreporting of symptoms due to patient and clinician expectations. The authors should clarify their rationale for including open-label trials and discuss whether any sensitivity analyses are planned to assess their impact on the findings.

2. Confounding Effects of Psychiatric Diagnoses

• The protocol includes patients with bipolar disorder and major depressive disorder, both of which can independently influence sexual function. While the authors plan subgroup analyses based on concomitant antidepressant use, it is unclear whether stratification or adjustments for psychiatric diagnoses are planned. Clarifying how the study will account for these confounding effects would strengthen the methodology.

3. Consistency in Study Inclusion Criteria

• The description of included study designs differs between the Abstract and Methods sections. In the Abstract, the authors state that both “controlled and uncontrolled randomized trials” will be included, while in the Methods, they specify “only blinded and open randomized controlled trials.” If the intention is to include only RCTs, the wording in the Abstract should be clarified to avoid potential confusion regarding whether non-controlled trials are eligible.

Reviewer #2: You have obtained many results. Write a conclusion based on these results, in a broad manner.

Thank you

Reviewer #3: Introduction:

- Clarify how the current review differs from the 2011 meta-analysis in terms of scope, the inclusion of newer antipsychotic drugs, and updated research findings.

- Include a clear, operational definition of "sexual dysfunction," as it is central to the focus of the review.

- Report available prevalence statistics of sexual dysfunction among patients taking antipsychotics to highlight its clinical importance.

- Provide brief explanations of terms such as hyperprolactinemia and dopamine antagonists to ensure accessibility for interdisciplinary readers, especially those from behavioral sciences and therapy-based backgrounds.

Methodology:

- Define “primary mental health disorder” with an operational definition and clarify which psychiatric conditions are included. Although diagnoses are mentioned, it is not clearly specified what is meant by “primary mental health disorder” in the context of the study.

- Specify whether the study accounts for the duration of antipsychotic use in inclusion or exclusion criteria.

- Indicate if the study considers or stratifies participants based on gender, given its relevance to sexual dysfunction.

- Clearly define the inclusion and exclusion criteria in terms of age, gender, and specific diagnoses.

- Justify the inclusion of serum prolactin as an outcome variable and explain its role as a physiological correlate of sexual dysfunction.

Exclusion Criteria:

- Clarify whether neurological conditions, substance use disorders, hypogonadism, and androgen disorders are explicitly excluded, and if they are not already captured under "somatic diseases."

Search Strategy:

- Mention if MeSH terms will be used in the systematic search strategy to ensure thorough and standardized literature retrieval.

Data Extraction and Analysis:

- Recommend including variables such as sample size per study, treatment or follow-up duration, and sex/gender distribution in the descriptive statistics to enhance the comprehensiveness of the analysis.

Discussion:

- Strengthen the rationale by clearly linking how the findings could influence clinical decisions, improve adherence to antipsychotic treatment, and guide future intervention strategies.

Reviewer #4: The submitted manuscript outlines a protocol for a systematic review and network meta-analysis investigating the prevalence and characteristics of sexual dysfunction associated with antipsychotic medication use.

The review plans to assess both the overall prevalence and specific domains (desire, arousal, orgasm), as well as the role of prolactin levels. The protocol is methodologically well-structured, with PROSPERO registration, PRISMA-P adherence, and use of rob 2 and GRADE frameworks, ensuring transparency and robustness. This topic is highly relevant due to the significant clinical impact of sexual side effects on treatment adherence, quality of life, and patient well-being.

The manuscript is clear in its aim, methodology, and expected outcomes. However, several key areas could be strengthened to enhance its methodological rigor, analytical depth, and clinical relevance.

1.Inclusion Criteria and Data Sources

The protocol excludes studies using semi-structured interviews or non-validated tools, focusing only on standardized questionnaires. While this increases internal validity, it risks excluding clinically meaningful data, especially qualitative insights.

Recommendation: Consider adding a sensitivity analysis including studies with semi-structured interviews or discuss this limitation more explicitly. You might also explore incorporating grey literature systematically (e.g., OpenGrey, ProQuest Dissertations) to mitigate publication bias.

2.Translation of Non-English Articles – Line 172-213

The authors plan to use ChatGPT 4.0 for machine translation of articles. This raises concerns about translation accuracy and loss of nuance.

Recommendation: Provide a clear quality control plan, e.g., using back-translation or secondary human verification, especially for key methodological sections.

3.Handling Missing Data

The protocol outlines the use of Furukawas method for imputing response rates, which is valid under certain assumptions.

Recommendation: Clarify the assumptions under which this method will be applied and plan sensitivity analyses to test robustness with and without imputed data.

4.Transitivity and Effect Modifiers in NMA

Although the authors mention assessing transitivity, the specific clinical and methodological effect modifiers are not clearly listed.

Recommendation: Explicitly describe variables (e.g., baseline sexual function, diagnosis, age, gender, medications, dosage) that will be evaluated to ensure comparability across trials.

5.Subgroup Analyses and Meta-Regression

Subgroup and metaregression analyses are briefly mentioned.

Recommendation: Specify the exact variables to be included, criteria for inclusion (e.g., minimum number of studies), and how multicollinearity will be managed.

6.Potential Bias and Publication Bias

While funnel plots and selection models are planned, no mention is made of bias introduced by industry sponsorship or selective reporting.

Recommendation: Include plans to evaluate sponsorship bias explicitly and consider including funding sources as a moderator in metaregression.

7.Implementation in Clinical Practice

The discussion touches on shared decision-making but lacks details on how results will translate to practice.

Recommendation: Expand the discussion on how the findings could inform treatment guidelines, shared decision-making tools, or clinical decision support systems.

Minor Comments

1.Typographical Errors

Line 36: "prolactin levels land sexual dysfunction" should be "and sexual dysfunction."

Multiple minor grammatical errors (e.g., "Antipsychotics are thought affect" should be "thought to affect").

Recommendation: A thorough language check is recommended to improve clarity.

2.Figure and Supplementary Materials

Ensure that Figure 1 (decision-tree) and supplementary files (search strategy, PRISMA-P checklist) are well-integrated and referenced properly in the main text.

3.References

The reference list is up-to-date and balanced; however, maybe these could enrich the discussion:

Clayton, A. H., & Montejo, A. L. (2014). Major depressive disorder, antidepressants, and sexual dysfunction. Journal of Clinical Psychiatry, 75(4), 383–391. https://doi.org/10.4088/JCP.13com13629

and this

Ebrahim, S., Akl, E. A., Mustafa, R. A., Sun, X., Walter, S. D., & Heels-Ansdell, D. (2013). Addressing continuous data measured with different instruments for outcome synthesis in systematic reviews: A survey of current practice and methods. Systematic Reviews, 2, Article 70. https://doi.org/10.1186/2046-4053-2-70

7. PLOS authors have the option to publish the peer review history of their article (what does this mean? ). If published, this will include your full peer review and any attached files.

**Do you want your identity to be public for this peer review?** For information about this choice, including consent withdrawal, please see our Privacy Policy .

Reviewer #1: No

Reviewer #2: No

Reviewer #3: No

Reviewer #4: No

---

## [Author Response · Author response to Decision Letter 1]

19 Jun 2025

We thank the reviewers for their valuable comments and suggestions. Please find our detailed responses and revisions in the accompanying rebuttal letter.

---

## [Decision Letter · Decision Letter 1]

1 Jul 2025

PONE-D-25-05627R1Sexual difficulty related to use of antipsychotics: a protocol for systematic review and meta-analysisPLOS ONE

Dear Dr. Herder,

Thank you for submitting your manuscript to PLOS ONE. After careful consideration, we feel that it has merit but does not fully meet PLOS ONE’s publication criteria as it currently stands. Therefore, we invite you to submit a revised version of the manuscript that addresses the points raised during the review process.

We look forward to receiving your revised manuscript.

Kind regards,

Hidetaka Hamasaki

Academic Editor

PLOS ONE

Journal Requirements:

Reviewers' comments:

Reviewer's Responses to Questions

**Comments to the Author**

1. Does the manuscript provide a valid rationale for the proposed study, with clearly identified and justified research questions?

Reviewer #4: Yes

2. Is the protocol technically sound and planned in a manner that will lead to a meaningful outcome and allow testing the stated hypotheses?

Reviewer #4: Yes

3. Is the methodology feasible and described in sufficient detail to allow the work to be replicable?

Reviewer #4: Yes

4. Have the authors described where all data underlying the findings will be made available when the study is complete?

Reviewer #4: Yes

5. Is the manuscript presented in an intelligible fashion and written in standard English?

Reviewer #4: Yes

6. Review Comments to the Author

You may also provide optional suggestions and comments to authors that they might find helpful in planning their study.

Reviewer #4: Thank you for thoroughly addressing previous major revision requests. The current manuscript is a substantial improvement, with clearer inclusion/exclusion criteria (particularly regarding the focus on validated sexual dysfunction questionnaires), an expanded and rigorous statistical analysis plan, and transparent plans for bias and transitivity assessments in NMA.

Specific strengths:

- The inclusion of a broad set of antipsychotics, including newer agents like aripiprazole, brexpiprazole, and cariprazine, which is clinically relevant.

- Comprehensive strategy for subgroup and sensitivity analyses, which increases the robustness of findings.

- The plan to handle language barriers in studies using machine translation followed by verification demonstrates thoughtful planning.

Minor recommendations:

- Ensure consistent use of terms: e.g., there are instances of “sexual difficulties” vs. “sexual dysfunction”, using one term throughout would improve clarity.

- In the introduction, the authors mention “sexual difficulties” affecting the “specific phases of sexual function”; consider clarifying early that this refers to desire, arousal, and orgasm to set context.

- Double-check minor typographical issues like “land sexual dysfunction” in the abstract (should be “and sexual dysfunction”).

Overall, I believe the manuscript is now technically sound, methodologically rigorous, and suitable for publication upon addressing these minor points.

7. PLOS authors have the option to publish the peer review history of their article (what does this mean? ). If published, this will include your full peer review and any attached files.

**Do you want your identity to be public for this peer review?** For information about this choice, including consent withdrawal, please see our Privacy Policy .

Reviewer #4: No

---

## [Author Response · Author response to Decision Letter 2]

4 Jul 2025

See attachment 'Response to reviewers (second revision)'

---

## [Decision Letter · Decision Letter 2]

18 Jul 2025

Sexual dysfunctions related to use of antipsychotics: a protocol for systematic review and meta-analysis

PONE-D-25-05627R2

Dear Dr. Herder,

We’re pleased to inform you that your manuscript has been judged scientifically suitable for publication and will be formally accepted for publication once it meets all outstanding technical requirements.

Kind regards,

Hidetaka Hamasaki

Academic Editor

PLOS ONE

Additional Editor Comments (optional):

Reviewers' comments:

Reviewer's Responses to Questions

**Comments to the Author**

1. Does the manuscript provide a valid rationale for the proposed study, with clearly identified and justified research questions?

Reviewer #4: Yes

2. Is the protocol technically sound and planned in a manner that will lead to a meaningful outcome and allow testing the stated hypotheses?

Reviewer #4: Yes

3. Is the methodology feasible and described in sufficient detail to allow the work to be replicable?

Reviewer #4: Yes

4. Have the authors described where all data underlying the findings will be made available when the study is complete?

Reviewer #4: Yes

5. Is the manuscript presented in an intelligible fashion and written in standard English?

Reviewer #4: Yes

6. Review Comments to the Author

You may also provide optional suggestions and comments to authors that they might find helpful in planning their study.

Reviewer #4: The revised protocol is well constructed, clearly presented, and methodologically rigorous. The authors have responded adequately to prior reviewer suggestions, including clarifying their statistical analysis strategy and enhancing discussion of sex/gender considerations.

Some optional suggestions to consider in final planning:

- In the limitations section, you may wish to explicitly reflect on the expected heterogeneity of reporting sexual dysfunction across studies (e.g., variability in definitions and outcome measures).

- Consider making the plan for narrative synthesis more concrete, especially in case meta-analysis proves infeasible due to study heterogeneity.

- If possible, include an example search string in the supplementary material to enhance reproducibility for future researchers.

Overall, this protocol meets high methodological standards and addresses an important clinical issue. I look forward to seeing the findings.

7. PLOS authors have the option to publish the peer review history of their article (what does this mean? ). If published, this will include your full peer review and any attached files.

**Do you want your identity to be public for this peer review?** For information about this choice, including consent withdrawal, please see our Privacy Policy .

Reviewer #4: No

---

## [Editor Report · Acceptance letter]

PONE-D-25-05627R2

PLOS ONE

Dear Dr. Herder,

I'm pleased to inform you that your manuscript has been deemed suitable for publication in PLOS ONE. Congratulations! Your manuscript is now being handed over to our production team.

Kind regards,

on behalf of

Dr. Hidetaka Hamasaki

Academic Editor

PLOS ONE